The neuroprotective effect of nicotine in Parkinson’s disease models is associated with inhibiting PARP-1 and caspase-3 cleavage

Lu Justin Y.D. 1
Su Ping 1
Barber James E.M. 2
Nash Joanne E. 2
Le Anh D. 1 3
Liu Fang 1 4
Wong Albert H.C. albert.wong@utoronto.ca 1 3 4
1 Campbell Family Mental Health Research Institute, Centre for Addiction and Mental Health , Toronto , Ontario , Canada
2 Centre for the Neurobiology of Stress, Department of Biological Sciences, University of Toronto, Scarborough , Toronto , Ontario , Canada
3 Department of Pharmacology and Toxicology, University of Toronto , Toronto , Ontario , Canada
4 Department of Psychiatry, University of Toronto , Toronto , Ontario , Canada
Rocha Joao
Electronic publication date: 2017 Oct 19
Publication date: 2017
Volume: 5
Electronic Location ID: e3933
Received 2017 May 24; Accepted 2017 Sep 26
Copyright: ©2017 Lu et al.
Copyright year: 2017
Copyright holder: Lu et al.
License: This is an open access article distributed under the terms of the Creative Commons Attribution License, which permits unrestricted use, distribution, reproduction and adaptation in any medium and for any purpose provided that it is properly attributed. For attribution, the original author(s), title, publication source (PeerJ) and either DOI or URL of the article must be cited.
License URL: https://creativecommons.org/licenses/by/4.0/

Keywords: Parkinson’s disease, Nicotine, Smoking, MPP+, 6-OHDA, Mouse, PARP-1, Caspase-3, Neuroprotection

Funding: The authors received no funding for this work.

==============================
Clinical evidence points to neuroprotective effects of smoking in Parkinson’s disease (PD), but the molecular mechanisms remain unclear. We investigated the pharmacological pathways involved in these neuroprotective effects, which could provide novel ideas for developing targeted neuroprotective treatments for PD. We used the ETC complex I inhibitor methylpyridinium ion (MPP+) to induce cell death in SH-SY5Y cells as a cellular model for PD and found that nicotine inhibits cell death. Using choline as a nicotinic acetylcholine receptor (nAChR) agonist, we found that nAChR stimulation was sufficient to protect SH-SY5Y cells against cell death from MPP+. Blocking α7 nAChR with methyllycaconitine (MLA) prevented the protective effects of nicotine, demonstrating that these receptors are necessary for the neuroprotective effects of nicotine. The neuroprotective effect of nicotine involves other pathways relevant to PD. Cleaved Poly (ADP-ribose) polymerase-1 (PARP-1) and cleaved caspase-3 were decreased by nicotine in 6-hydroxydopamine (6-OHDA) lesioned mice and in MPP+-treated SH-SY5Y cells. In conclusion, our data indicate that nicotine likely exerts neuroprotective effects in PD through the α7 nAChR and downstream pathways including PARP-1 and caspase-3. This knowledge could be pursued in future research to develop neuroprotective treatments for PD.

Introduction

Parkinson’s disease (PD) is a neurodegenerative disorder affecting the nigrostriatal dopamine tract that regulates the initiation and fluency of voluntary movement. Patients present with a characteristic set of neurological symptoms that include tremor, muscle rigidity, bradykinesia, stooped posture, shuffling gait and a lack of facial expression (Magrinelli et al., 2016). In the advanced stages, patients may also develop a subcortical dementia and a variety of neuropsychiatric symptoms (Sveinbjornsdottir, 2016). Symptoms can be alleviated temporarily with L-dopa and carbidopa but this does not alter the progression of the illness or the death of nigrostriatal neurons (Connolly & Lang, 2014).

Although smoking cigarettes has well-documented adverse health effects including lung cancer and cardiovascular disease, smokers are less likely to develop PD (Ascherio & Schwarzschild, 2016; Baron, 1996; Breckenridge et al., 2016; Hernan et al., 2002; Polito, Greco & Seripa, 2016; Ritz et al., 2007). This is also true for passive exposure to second-hand smoke (Searles Nielsen et al., 2012) or chewing tobacco (O’Reilly et al., 2005), and appears to be dose-dependent (Thacker et al., 2007). The mechanisms underlying the potential neuroprotective effects of tobacco exposure remain unclear, but hypotheses include: (1) interactions between the dopamine and acetylcholine neurotransmitter systems, (2) reduction of oxidative stress, (3) modulation of neuroinflammation, and (4) non-specific cognitive enhancing effects (Barreto, Iarkov & Moran, 2014). Although nicotine is the most well-known component of tobacco, cotinine and other metabolites may also play a role (Barreto, Iarkov & Moran, 2014).

Animal studies have provided useful insights into potential mechanisms for the protective effects of tobacco in PD. Lesioning cholinergic neurons in the pedunculopontine nucleus resulted in loss of substantia nigra dopaminergic neurons (Bensaid et al., 2016), showing that physiological levels of acetylcholine are required for survival of nigrostriatal dopaminergic neurons. In rats with 6-hydroxydopamine (6-OHDA) lesions of the medial forebrain bundle, nicotine or the α7 nAChR agonist ABT-107 improved neurological functioning in conjunction with restoring dopamine transporter levels and dopamine release (Bordia et al., 2015). Treatment with a different α7 nAChR agonist 3-[(2,4-dimethoxy)benzylidene]-anabaseine dihydrochloride (DMXBA) or nicotine also protects dopamine neurons in rats injected with 6-OHDA (Costa, Abin-Carriquiry & Dajas, 2001; Suzuki et al., 2013). Blocking nAChR with chlorisondamine prevents the protective effects of nicotine in vivo. Another common animal model for PD relies on 1-methyl-4-phenyl-1,2,3,6-tetrahydropyridine (MPTP) that is toxic to nigrostriatal dopamine neurons. Both cigarette smoke and nicotine increased the survival of these neurons in MPTP mice (Parain et al., 2003). Similar results have been reported in non-human primates exposed to MPTP (Quik et al., 2006), and in mice models of PD using methamphetamine to induce dopamine neuron toxicity (Maggio et al., 1997). Nicotine increases the levels of fibroblast growth factor-2 (FGF-2) and brain-derived neurotrophic factor in rat striatum in these models, which could be one mechanism for neuroprotection (Mudo et al., 2007).

Cellular model systems have also been used to investigate specific pathways through which nicotine and other tobacco constituents could protect neurons in PD. Using cultured mouse ventral midbrain neurons that included dopamine neurons, one group used tunicamycin as an endoplasmic reticulum stressor and found that nicotine, at levels comparable to those achieved through smoking cigarettes, attenuated the unfolded protein response (Srinivasan et al., 2016). There is also evidence that dopamine release can be regulated by presynaptic nAChR in rat brain slices (Giorguieff-Chesselet et al., 1979), and mouse striatal synaptosomes (Grady et al., 1992; Rapier, Lunt & Wonnacott, 1990).

Poly (ADP-ribose) polymerase-1 (PARP-1) and caspase have both been implicated in the pathophysiology or etiology of PD. PARP-1 is a DNA-damage sensor that is activated in some PD models such as the MPTP mouse (Wang et al., 2003), and inhibiting PARP-1 reduced dopamine neuron death from MPTP (Iwashita et al., 2004), alpha synuclein and MPP+ (Outeiro et al., 2007). PARP-1 also mediates dopamine neuron degeneration in the 6-OHDA mouse PD model (Kim et al., 2013). Mutations in PARP-1 protect against mitochondrial dysfunction and neurodegeneration in mouse models of PD with mutations in the Parkin gene (Lehmann et al., 2016), and in human clinical populations (Infante et al., 2007).

Caspase-3 has been implicated in cleavage of a proapoptotic kinase protein kinase C delta (PKCdelta) that mediates neuron death in both MPP+ and 6-OHDA cellular PD models (Da Costa, Masliah & Checler, 2003; Kanthasamy et al., 2006; Shimoke & Chiba, 2001). There is also evidence that caspase-1 activation is the final step in apoptotic cell death in PD (Hartmann et al., 2000; Tatton, 2000). Acteoside binding to caspase-3 is neuroprotective in the rotenone rat PD model (Yuan et al., 2016), and caspase-3 activation has been observed to be important in a number of pathways related to PD (Shukla et al., 2014; Zawada et al., 2015). Genetic disruption of caspase-3 is also protective against the effects of MPTP (Yamada et al., 2010). To our knowledge, there have not been attempts to investigate whether the neuroprotective effects of nicotine involve PARP-1 or caspase.

In summary, there is evidence that nicotinic cholinergic drugs may delay progression of PD (Perez, 2015), and thus, the α7 nAChR has been proposed as a target for new medications to treat PD (Quik et al., 2015). However, since the mechanisms underlying the neuroprotective effects remain unclear, we sought to further investigate the role of the α7 nicotinic acetylcholine receptor (α7 nAChR) in mediating the protective effects of nicotine in PD. We used the ETC complex I inhibitor methylpyridinium ion (MPP+) to induce cell death in SH-SY5Y cells as a cellular model for PD and used 6-hydroxydopamine (6-OHDA) lesions as a mouse model for PD. We investigated the potential involvement of PD-related molecules PARP-1 and caspase in both of these model systems.

Materials and Methods

Cell culture and treatment

SH-SY5Y cells are derived from a human neuroblastoma and are often used as a cellular model for PD because they express tyrosine hydroxylase, dopamine-beta-hydroxylase, and the dopamine transporter. Xie, Hu & Li (2010) SH-SY5Y cells (American Type Culture Collection (ATCC), Manassas, VA) were maintained as a monolayer in Dulbecco’s Modified Eagle Medium (DMEM) (Gibco, ON, Canada) with 10% fetal bovine serum (Gibco, ON, Canada), 100 U/ml penicillin (Sigma-Aldrich, Oakville, ON, Canada), and 100 U/ml streptomycin (Sigma-Aldrich, Oakville, ON, Canada). Cells were cultured in a humidified atmosphere of 5% CO2, at 37 °C. All cells were cultured in 100-mm (diameter) cell culture plates (BD Biosciences, ON, Canada) until ∼80% confluence and then seeded into 24-well plates (BD Bioscience, ON, Canada) to achieve ∼90% confluence 24–28 h prior to treatment. The medium was replaced by DMEM without fetal bovine serum 12 h before treatments.

Drugs

MPP+ (methylpyridinium ion) was purchased as MPP+ iodide from Sigma-Aldrich, dissolved in water to a stock concentration of 500 mM, and wrapped with foil to protect from light. Choline, nicotine and methyllycaconitine (MLA) were purchased from Tocris Bioscience. Nicotine was used at a concentration of 2 mM for in vitro experiments based on previous reports (Ke et al., 1998; Wang et al., 2011). We used MLA at a concentration of 20 µM based on a previous report that MLA at 5 µM and 10 µM could alleviate amyloid- β peptide-induced cytotoxicity in SH-SY5Y cells, without affecting cell viability (Zheng et al., 2014). At 20 µM, MLA could theoretically interact with α4β2 and α6β2 receptors, but no α4 and α6 receptor subunit mRNA was detected in SH-SY5Y cells (Gould et al., 1992; Lukas, Norman & Lucero, 1993). The α7 acetylcholine receptor subunit has good expression levels in SH-SY5Y cells (Peng et al., 1994).

Propidium iodide (PI) and Hoechst33342 staining

Cultured SH-SY5Y cells were gently rinsed with phosphate-buffered saline (PBS) (pre-warmed in 37 °C) twice, incubated with 50 μg/ml PI (Invitrogen, Carlsbad, CA) or double labeling with Hoechst 33342 (20 μg/ml) (Invitrogen, Carlsbad, CA) and PI for 30 min, and then rinsed three times with PBS. Fluorescent intensity was measured by a plate reader (Victor 3; Pekin-Elmer, Waltham, MA). The level of cell death was defined as the ratio of PI: Hoechst 33342. The fraction of dead cells was normalized to the cell toxicity that occurred in the control group.

Protein extraction

Striatial tissues were dissected from mice in 6-OHDA exposure models. Striata were homogenized in ice cold buffer containing (in mmol/L): 50 Tris-Cl, pH 7.4, 150 NaCl, 2 EDTA, 1 PMSF plus 1% Igepal CA-630, 0.5–1% sodium deoxycholate, 1% Triton X-100 and protease inhibitor mixture (5  μL/100 mg of tissue; Sigma-Aldrich, Okaville, ON, Canada) on ice and shaken at 4 °C for 1 h. Striatal tissues dissolved in the lysis buffer was centrifuged at 12,000 g for 10 min at 4 °C to yield the total protein extract in the supernatant. The concentration of protein was measured with the BCA protein assay kit (Pierce Protein Biology, ON, Canada). Equal amounts of samples (50∼100 μg) were denatured and subjected to 10% SDS-PAGE and Western blot analyses.

Gel electrophoresis and Western blot analyses

Samples were separated using SDS-PAGE with 10% separating gel and 5% stacking gel, and transferred to a nitrocellulose membrane after gel electrophoresis. After blocking for 1 h with 5% fat-free milk powder in TBST (10 mM Tris, 150 mM NaCl, 0.05% Tween-20, pH7.4), blots were incubated overnight at 4 °C with primary antibodies: 1:200 anti-PARP-1 (Santa Cruz Biotechnology, Dallas, Texas), 1:10,000 anti- α-Tubulin (Sigma-Aldrich) and 1:200 anti-caspase-3 (Santa Cruz Biotechnology, Dallas, TX, USA). After washes, blots were incubated with HRP-conjugated secondary antibodies (Sigma-Aldrich, Okaville, ON, Canada) for 2 h at room temperature. Immunoactivity was visualized with ECL Western blot detection reagents (GE Healthcare, Little Chalfont, UK). Data representative of three experimental replicates are shown.

Unilateral 6-OHDA lesions and nicotine administration

The animal studies were approved by the University Animal Care Committee (UACC) at the University of Toronto in accordance with the Canadian Council on Animal Care (CCAC) guidelines (IRB approval number 20010879). Surgeries were performed as previously described (Thiele et al., 2011; Thiele, Warre & Nash, 2012). In brief, 30 min prior to surgery, a mixture of desipramine hydrochloride (25 mg/kg; Sigma Aldrich) and pargyline hydrochloride (5 mg/kg; Sigma Aldrich) in 0.9% sterile saline (pH 7.4) was systemically administered intra-peritoneally (i.p.). C57Bl/6J mice (P35, 24–28 g) were anaesthetised (isoflurane (Abbott), 2–3%) and placed in a stereotaxic frame (David Kopf Instruments, USA). 6-hydroxydopamine (6-OHDA) (15 μg/ μl, 0.02% ascorbic acid, w/v in 0.9% saline) or vehicle was unilaterally injected into the medial forebrain bundle (MFB) at a rate of 0.1 μl/min (total delivery of 3 μg total, as a 0.2 μl bolus) at the following coordinates: AP: −1.2 mm, ML: −1.1 mm, and DV: −5.0 mm (Paxinos & Franklin, 2007). This protocol results in a >95% dopamine depletion of the SNc (Thiele et al., 2011; Thiele, Warre & Nash, 2012).

Seven days prior to 6-OHDA lesion surgeries, animals were given nicotine or saline control by subcutaneous injection (s.c.) (MP Biomedicals, LLC, Santa Ana, CA, USA) twice daily for two weeks. For the first three days animals received a dose of 0.4 mg/kg, which was then increased to 0.8 mg/kg for four days prior to surgery. This dose was continued for one week post-surgery until subjects were sacrificed for tissue collection.

Statistical analysis

Levene’s homogeneity test or F test was used to compare the variances between groups. For equal variances, data were analyzed either by t-test, one-way analysis of variance (ANOVA) followed by Tukey’s test, or two-way analysis of variance (ANOVA) followed by Bonferroni or Tukey’s post-tests (SPSS Statistics, I.B.M Corporation, USA). For groups with unequal variance, data were analyzed either with a t-test with Welch’s correction, a one-way analysis of variance (ANOVA), or two-way ANOVA, followed by Dunnett’s post hoc test. Data are expressed as mean ± standard error of mean (SEM). The significance levels of p < 0.05, p < 0.01, or p < 0.001 were used for all analyses.

Figure 1 Nicotine protects SH-SY5Y cells against MPP+-induced cell death.

(A) MPP+ treatment (500 µM, 24 hrs) in SH-SY5Y cells increased the level of cell death, as compared to control cells. *p < 0.05 as compared to those in control group, n = 5, t-test. (B) Pre-treatment with nicotine (2 mM, 30 min) prior to MPP+ exposure in SH-SY5Y cells decreased the level of cell death as compared to those treated with MPP+ only. *p < 0.05 as compared to those of control group, ##p < 0.01 as compared to MPP+ group, n = 5 for control and MPP+ groups, n = 3 for nicotine and MPP+ with nicotine groups, two-way ANOVA followed by Bonferroni post-tests. All data are shown as mean ± SEM. The level of cell death was detected using PI (50 µg/ml) and Hoechst33342 (20 µg/ml) double staining, and was defined as the ratio of fluorescent intensity of PI: Hoechst33342.

Results

Nicotine inhibits MPP+-induced SH-SY5Y cell death

Tobacco exposure is associated with decreased risk for PD (O’Reilly et al., 2005; Ritz et al., 2007; Searles Nielsen et al., 2012) and nicotine is the most prominent psychoactive component of tobacco. Thus, we first investigated if nicotine could protect against cell death in a cellular model of PD: MPP+-induced SH-SY5Y cell death. As shown in Fig. 1A, using propidium iodide (PI) staining, MPP+ treatment (500 μM, 24 h) induced more SH-SY5Y cell death compared to control cells (control: 1.00 ± 0.099; MPP+: 1.40 ± 0.086). Pre-treatment with nicotine (2 mM, 30 min) prior to MPP+ treatment, decreased the level of cell death, as compared cells treated with MPP+ alone (MPP+: 1.40 ± 0.086; MPP+ with nicotine: 0.88 ± 0.068; Fig. 1B). These data show that nicotine can inhibit MPP+-induced SH-SY5Y cell death.

nAChR is involved in the protective effect of nicotine

To determine whether the protective effects of nicotine are mediated by activation of the nAChR, we investigated if activation of nAChR without using nicotine, inhibits MPP+-induced SH-SY5Y cell death. Nicotine is an nAChR agonist and previous studies have shown that other nAChR agonists can protect against nigrostriatal dopamine neuron damage in PD animal models (Dajas et al., 2001; Janson et al., 1988; Maggio et al., 1998; Mudo et al., 2007). As shown in Fig. 2, choline (1 mM, 30 min), a nAChR specific agonist, decreased the level of cell death, when administered prior to MPP+ treatment (Control: 1.00 ± 0.099; Choline: 0.98 ± 0.10; MPP+: 1.40 ± 0.086; MPP+ with choline: 0.81 ± 0.12). These results indicate that activation of nAChR prevents SH-SY5Y cells from MPP+-induced cell death and suggest that nAChR activation is sufficient to protect SH-SY5Y cells against MPP+-induced death.

Figure 2 Choline protects SH-SY5Y cells against MPP+-induced cell death.

Pre-treatment with choline (1 mM, 30 min), a specific nAChR agonist, followed by MPP+ treatment (500 µM, 24 hrs) in SH-SY5Y cells decreased the level of cell death as compared to those treated with MPP+ only. *p < 0.05 as compared to those of control group, ###p < 0.001 as compared to MPP+ group, n = 5, two-way ANOVA followed by Bonferroni post-tests. All data are shown as mean ± SEM. The level of cell death was detected using PI and Hoechst33342 double staining, and was defined as the ratio of fluorescent intensity of PI: Hoechst33342.

Figure 3 Blockade of α7 nAChR inhibits the protective effect of nicotine and choline against MPP+-induced SH-SY5Y cell death.

(A) MLA (20 µM, 30 min), a specific antagonist of α 7 nAChR, increased the level of cell death when administered prior to nicotine (2 mM, 30 min) and MPP+ (500 µM, 24 hrs) treatments in SH-SY5Y cells, as compared to those treated with nicotine and MPP+ alone. ***p < 0.001 as compared to those of control group, #p < 0.05 as compared to MPP+ group, n = 4, one-way ANOVA followed by Tukey’s test. (B) MLA (20 µM, 30 min), a specific antagonist of α7 nAChR, increased the level of SH-SY5Y cell death when administered prior to choline (1 mM, 30 min) and MPP+ (500 µM, 24 hrs) treatments, as compared to those treated with choline and MPP+ alone. ***p < 0.001 compared to the control group, #p < 0.05 as compared to the MPP + group, n = 4, one-way ANOVA followed by Tukey’s test. All data are shown as mean ± SEM. The level of cell death was detected using PI and Hoechst33342 double staining, and was defined as the ratio of fluorescent intensity of PI: Hoechst33342.

α7 nAChR mediates the protective effect of nicotine against MPP+-induced SH-SY5Y cell death

We hypothesized that the α7 nAChR could be the receptor through which nicotine has neuroprotective effects in PD. Activation of the α7 nAChR has protective effects in other neurodegenerative disorders, and in Alzheimer’s disease models (Fan, Gu & Wei, 2015; Hu et al., 2015; Shen & Wu, 2015). To confirm if α7 nAChR mediates the effect of nicotine to protect against MPP+-induced cell death, we pre-treated SH-SY5Y cells with methyllycaconitine (MLA) (20 μM, 30 min), a α7 nAChR-specific antagonist, followed by MPP+ as above and either nicotine (2 mM, 30 min) or choline (1 mM, 30 min). As shown in Fig. 3A, MLA treatment increased the level of cell death when administered prior to nicotine and MPP+ treatments, as compared to those treated with nicotine and MPP+ alone (control: 1.00 ± 0.053; MPP+: 1.70 ± 0.119; MPP+ with nicotine: 1.37 ± 0.0351; MPP+ with nicotine and MLA: 1.81 ± 0.0628). Similarly, Fig. 3B shows that choline can reduce the cell death induced by MPP+ and this effect is blocked by MLA (Control: 1.00 ± 0.053; MPP+: 1.70 ± 0.119; MPP+ with choline: 1.34 ± 0.0197; MPP+ with choline and MLA: 1.85 ± 0.0796). These data indicate that α7 nAChR signaling is necessary for the neuroprotective effect of nicotine.

Figure 4 Nicotine pre-treatment inhibits PARP-1 and caspase-3 cleavage in MPP+-treated SH-SY5Y neuroblastoma cells.

(A) Western blot analysis showing that cleaved caspase-3 decreased in SH-SY5Y cells pre-treated with nicotine before MPP+ treatment, as compared to those treated with MPP+ only. α-Tubulin was used as a loading control. (B and C) Densitometric analysis of expression levels of full length (B) and cleaved (C) caspase-3. The expression level of caspase-3 was defined as the ratio of the intensity of caspase-3: α-Tubulin, and was normalized as percentage of the control group, **p < 0.01, n = 3, two-way ANOVA. (D) Western blot analysis showing that cleaved PARP-1 was decreased in SH-SY5Y cells pre-treated with nicotine before MPP+ treatment, as compared to those treated with MPP+ only. α-Tubulin was used as a loading control. (E and F) Densitometric analysis of the intensity of expression levels of full length (E) and cleaved (F) PARP-1. The expression level of PARP-1 was defined as the ratio of the intensity of PARP-1: α-Tubulin, and was normalized as percentage of the control group, ***p < 0.001, n = 3, two-way ANOVA. All data are shown as mean ± SEM.

Nicotine pre-treatment inhibits PARP-1 and caspase-3 cleavage in MPP+-treated SH-SY5Y neuroblastoma cells

To investigate potential mechanisms mediating the effect of nicotine in the MPP+ SH-SY5Y cellular model of PD, we examined PARP-1 and caspase-3 cleavage with and without nicotine. As shown in Fig. 4, nicotine pre-treatment inhibits the cleavage of caspase-3 (Control: 1.00 ± 0; nicotine: 1.05 ± 0.078; MPP+: 1.27 ± 0.026; MPP+ with nicotine: 0.933 ± 0.073) and PARP-1 (Control: 1.00 ± 0; nicotine: 0.919 ± 0.054; MPP+: 1.17 ± 0.022; MPP+ with nicotine: 0.919 ± 0.022), compared to the control cells, using α-Tubulin as the loading control against which the other proteins were normalized.

The neuroprotective effect of nicotine is associated with decreased PARP-1 and caspase-3 cleavage

To expand on the in vitro results above, we performed unilateral 6-hydroxydopamine lesions in mice as an in vivo model of PD. We first confirmed that the 6-OHDA lesion was causing the expected death of dopamine neurons and that nicotine had a neuroprotective effect in vivo, by measuring the amount of tyrosine hydroxylase as a proxy for dopamine neuron survival. Tyrosine hydroxylase is the rate limiting enzyme in the synthesis of catecholamines including dopamine. Figure 5 shows that the lesioned hemisphere of the brain has lost approximately half of the TH-containing neurons, while nicotine treatment protected almost all of these neurons from death (non-lesioned side in 6-OHDA mice: 1.00 ± 0.070; lesioned side in 6-OHDA mice: 0.49 ±0.19, Fig. 5B; and non-lesioned side in 6-OHDA mice with nicotine: 1.00 ± 0.0473; lesioned side in 6-OHDA mice with nicotine: 0.978 ± 0.0565, Fig. 5C).

To investigate potential mechanisms underlying this neuroprotective effect of nicotine, we measured the expression of Poly [ADP-ribose] polymerase 1(PARP-1) and caspase-3 using Western blots. We analyzed protein from solubilized striatal tissue of mice exposed to 6-hydroxydopamine (6-OHDA) with or without nicotine. Both cleaved PARP-1 (6-OHDA: 1.00 ± 0.54; Nicotine+6-OHDA: 0.740 ± 0.022; Figs. 6A–6C) and cleaved caspase-3 (6-OHDA: 1.00 ± 0.017; Nicotine+6-OHDA: 0.718 ± 0.053; Figs. 6D–6F) were decreased by nicotine pre-treatment in 6-OHDA mice pretreated with nicotine. This indicates that nicotine pre-treatment inhibits PARP-1 and caspase-3 cleavage in this PD mouse model. The main cleaved PARP-1 fragment was 89 KDa in size (full length 116 kDa).

We also performed a control experiment to examine whether nicotine alone might alter PARP-1 or caspase-3 cleavage, using Western blots to quantify the amount of the intact, full-length protein vs. the cleaved form, in the unlesioned hemisphere of 6-OHDA mice. As shown in Fig. 7, there is no significant effect of nicotine alone on the cleavage of these two proteins. These data demonstrate that the neuroprotective effect of nicotine for dopamine neurons in PD models is associated with PARP-1 and caspase-3 cleavage pathways.

Figure 5 Tyrosine Hydroxylase expression level decreased in lesioned hemisphere of 6-OHDA mouse model of PD.

(A) Western blot analysis showing that tyrosine hydroxylase expression level decreased in striatal tissues of lesioned hemisphere from mice exposed to 6-OHDA only, but did not change in mice pretreated with nicotine before 6-OHDA exposure. α-Tubulin was used as a loading control. (B) Densitometric analysis of expression levels of tyrosine hydroxylase. The expression level of tyrosine hydroxylase was defined as the ratio of the intensity of tyrosine hydroxylase: α-Tubulin, and was normalized as percentage of non-lesioned hemisphere exposed to 6-OHDA only, n = 3, two-way ANOVA. All data are shown as mean ± SEM. *p < 0.05 as compared to Non-lesioned hemisphere in 6-OHDA mice.

Figure 6 Nicotine pre-treatment inhibits PARP-1 and caspase-3 cleavage in striatal tissue from 6-OHDA mouse model of PD.

(A) Western blot analysis showing that cleaved PARP-1 decreased in striatal tissues from mice pre-treated with nicotine before 6-OHDA exposure, as compared to those exposed to 6-OHDA only. α-Tubulin was used as a loading control. (B and C) Densitometric analysis of expression levels of full length (B) and cleaved (C) PARP-1. The expression level of PARP-1 was defined as the ratio of the intensity of PARP-1: α -Tubulin, and was normalized as percentage of the 6-OHDA group, *p < 0.05 compared to the 6-OHDA group, n = 3, t-test. (D) Western blot analysis showing that cleaved caspase-3 was decreased in striatal tissues from mice pre-treated with nicotine prior to 6-OHDA exposure, as compared to those exposed to 6-OHDA only. α-Tubulin was used as a loading control. (E and F) Densitometric analysis of the intensity of expression levels of full length (E) and cleaved (F) caspase-3. The expression level of caspase-3 was defined as the ratio of the intensity of caspase-3: α-Tubulin, and was normalized as percentage of the 6-OHDA group, **p < 0.01 compared to the 6-OHDA group, n = 3, t-test. All data are shown as mean ± SEM.

Figure 7 Nicotine pre-treatment does not change PARP-1 and caspase-3 cleavage in striatal tissue from non-lesioned hemisphere of 6-OHDA mouse model of PD.

(A) Western blot analysis showing no difference of cleaved caspase-3 in striatal tissues of non-lesioned hemisphere from mice pre-treated with nicotine before 6-OHDA exposure, as compared to those exposed to 6-OHDA only. α-Tubulin was used as a loading control. (B and C) Densitometric analysis of expression levels of full length (B) and cleaved (C) caspase-3. The expression level of caspase-3 was defined as the ratio of the intensity of caspase-3: α-Tubulin, and was normalized as percentage of the 6-OHDA group, n = 3, t-test. (D) Western blot analysis showing no difference of cleaved PARP-1 was decreased in striatal tissues of the non-lesioned hemisphere from mice pre-treated with nicotine prior to 6-OHDA exposure, as compared to those exposed to 6-OHDA only. α-Tubulin was used as a loading control. (E and F) Densitometric analysis of the intensity of expression levels of full length (E) and cleaved (F) PARP-1. The expression level of PARP-1 was defined as the ratio of the intensity of PARP-1: α-Tubulin, and was normalized as percentage of the 6-OHDA group, n = 3, t-test. All data are shown as mean ± SEM.

Discussion

The data presented above demonstrate that nicotine inhibits MPP+-induced SH-SY5Y cell death through activating α7 nAChR, and inhibits PARP-1 and caspase-3 cleavage in the 6-OHDA mouse model for PD. We first demonstrated that activation of nAChR with either nicotine or choline is sufficient to protect SH-SY5Y cells from MPP+ toxicity. Nicotine also inhibits PARP-1 and caspase-3 cleavage in MPP+-treated SH-SY5Y cells. We then showed that α7 nAChR activation is necessary for these neuroprotective effects by using the α7 nAChR antagonist methyllycaconitine, which reduces the number of cells rescued by nicotine. Finally, we used the in vivo 6-OHDA mouse model for PD to demonstrate that nicotine inhibits PARP-1 and caspase-3 cleavage, suggesting a potential downstream molecular mechanism for neuroprotection in PD.

This study provides additional knowledge of potential mechanisms to explain the clinical phenomenon of reduced PD incidence in smokers and other people exposed to tobacco. Some have suggested that people who become tobacco users may have an underlying trait that also renders them less susceptible to PD (Barreto, Iarkov & Moran, 2014), a form of “reverse causation” rather than nicotine actually being neuroprotective. One group found that PD patients are able to quit smoking more easily than matched population controls (Ritz et al., 2014). However, there are two main arguments against this interpretation. The first is that passive exposure to cigarette smoke is also associated with a dose-dependent decreased risk for PD (Searles Nielsen et al., 2012), and the second is the large body of data from experimental animal and cellular models (Barreto, Iarkov & Moran, 2014; Perez, 2015).

The nAChRs are obvious potential starting points for the mechanism of neuroprotection by nicotine in PD, and these receptors have been investigated in many other studies (Barreto, Iarkov & Moran, 2014; Perez, 2015; Quik et al., 2015). α7nAChR agonists are also likely to have significant impacts in PD via the regulation of the immune system and intestinal permeability (Anderson et al., 2016). However, our study is unique in using SH-SY5Y cells exposed to MPP+ as an in vitro model to investigate the neuroprotective effects of nicotine and nAChR activation. Also novel is our attempt to investigate the effect of nicotine on PARP-1 and caspase-3 cleavage in the 6-OHDA mouse model for PD. These elements provide insight into molecular mechanisms and potential targets for developing new PD treatments.

Our results do not exclude the involvement of other neuroprotective mechanisms. Several signaling pathways that promote cell survival are enhanced by stimulating nAChR, including the Src family-PI3 K-AKT pathway, with subsequent upregulation of Bcl-2 and Bcl-x, JAK2/STAT3 and MEK/ERK (Kawamata & Shimohama, 2011). Nicotine can protect SH-SY5Y cells from other types of insults, such as beta-amyloid toxicity, through Erk1/2-p38-JNK-dependent signaling pathways (Xue et al., 2014). However, there are no published studies investigating the role of PARP-1 or caspase in the neuroprotective effects of nicotine in PD disease models.

Caspases are a family of proteases that are activated during apopototic cell death (Kroemer & Martin, 2005). There are twelve numbered caspases, some of which initiate or execute apoptosis, but these enzymes also regulate inflammation and cell differentiation (Galluzzi et al., 2016). Caspases are initially synthesized as an inactive pro-caspase, which must undergo dimerization or oligomerization and then cleavage to become active (Shi, 2004). Caspases are involved in the pathophysiology of PD through mediating dopaminergic neuron death from MPTP (Furuya et al., 2004; Qiao et al., 2017; Viswanath et al., 2001), promoting synuclein aggregation (Wang et al., 2016), and cleaving Transactivation response DNA-binding protein 43 (TDP-43), which is a primary component of Lewy bodies in PD (Kokoulina & Rohn, 2010).

A number of neuroprotective compounds that have been studied in PD animal models also affect caspases, such as telmisartan (an angiotensin II type 1 receptor blocker) (Tong et al., 2016), and nerve growth factor (NGF) (Shimoke & Chiba, 2001). Directly blocking a caspase-3 cleavage site on the proapoptotic protein kinase C delta has neuroprotective effects in MPP+ and 6-OHDA PD models (Kanthasamy et al., 2006). Despite the prominence of caspases in neuronal death in PD, they may not be viable targets for treatment since directly blocking caspase-8 resulted in a switch from apoptosis to necrosis (Hartmann et al., 2001), and this may apply to other caspases as well (Kroemer & Martin, 2005). Modulating caspase function in PD through the nicotinic receptors may be a better approach for developing new treatments.

PARP-1 enzymes are involved in a number of neurodegenerative disorders including Alzheimer’s disease and PD (Martire, Mosca & d’Erme, 2015). PARP-1 has DNA binding domains that detect DNA damage and facilitate repair. When PARP-1 levels are too high or when DNA damage is too severe, cell death is initiated (Burkle, 2001), and this decision is regulated by NAD+ depletion (Alano et al., 2010). During cell death programs, PARP-1 is cleaved into fragments that are specific to different apoptotic pathways (Chaitanya, Steven & Babu, 2010). The 89 KDa fragment we detected appears during apoptosis, and could have been generated by the action of caspase-3, caspase-7 (Lazebnik et al., 1994) or the lysosomal proteases cathepsin B or D (Gobeil et al., 2001).

Some PARP-1 genetic variants are protective against PD (Infante et al., 2007), and the involvement of PARP-1 in PD pathophysiology includes regulation of alpha-synuclein expression (Chiba-Falek et al., 2005), and modification of p53 in the MPTP model (Mandir et al., 2002). Small molecule inhibitors of PARP-1 reduce cell death induced by alpha-synuclein and MPP+ (Outeiro et al., 2007), consistent with our results above. PARP-1 induced depletion of NAD+ could also contribute to decreasing sirtuins and mitochondrial dysfunction in PD (Anderson & Maes, 2014).

In conclusion, we have demonstrated that the neuroprotective effects of nicotine in animal and cellular models of PD is mediated by activation of α7 nAChR and the inhibition of PARP-1 and caspase-3 cleavage. All of these molecules have been previously implicated in the pathophysiology of PD, but until now have not been linked together. This knowledge could be used to aid development of novel treatments for PD, but further work to delineate the molecular pathway linking α7 nAChR to PARP-1 and caspase-3 is required.

Supplemental Information

Supplemental Information 1 Raw data

Click here for additional data file.

Supplemental Information 2 Full images of blots that form the basis for Fig. 4

The set of four blots that are part of the supplementary data supporting Fig. 4.

Click here for additional data file.

Additional Information and Declarations

Competing Interests

Author Contributions

Animal Ethics

Data Availability

Albert H.C. Wong is an Academic Editor for PeerJ.

Justin Y.D. Lu performed the experiments, analyzed the data, prepared figures and/or tables, reviewed drafts of the paper.

Ping Su conceived and designed the experiments, performed the experiments, analyzed the data, prepared figures and/or tables, reviewed drafts of the paper.

James E.M. Barber and Joanne E. Nash contributed reagents/materials/analysis tools, reviewed drafts of the paper.

Anh D. Le conceived and designed the experiments, contributed reagents/materials/analysis tools, reviewed drafts of the paper.

Fang Liu and Albert H.C. Wong conceived and designed the experiments, wrote the paper, reviewed drafts of the paper.

The following information was supplied relating to ethical approvals (i.e., approving body and any reference numbers):

The animal studies were approved by the University Animal Care Committee (UACC) at the University of Toronto in accordance with the Canadian Council on Animal Care (CCAC) guidelines (IRB approval number 20010879).

The following information was supplied regarding data availability:

The raw data has been uploaded as a Supplemental File.

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
