# Peer review of "The neuroprotective effect of nicotine in Parkinson’s disease models is associated with inhibiting PARP-1 and caspase-3 cleavage"

_PeerJ, doi:10.7717/peerj.3933_

## Round 0.1 · original submission · Major Revisions

Your manuscript has been reviewed by two reviewers, who have raised several important points that you have to consider before a final decision can be made. Please review the manuscript detailing all the changes you will make to meet reviewers' critiques. Note that if you do not reply clearly to the points raised by them, the manuscript will be re-submitted for a second round of review.

Reviewer 1 ·

Basic reporting

The study: "The neuroprotective effect of nicotine in Parkin's disease models is associated with inhibiting PARP-1 and caspase-3 cleavage", propose to give evidence that nicotine administration can contrast cell death in the MPP+ cellular model for PD and increase PARP-1 and caspase-3 expression in the 6-OHDA mouse model for PD. They found that nicotinic cholinergic drugs may delete PD progression, in particular through α7 nAChR activation.
Besides some typos (Figure 1B in the legend, nicotine; line 231, ih), the article is written using a clear and professional English, and is supported by a sufficient literature. However, there are several concerns for me as follows:
1. the means and the standard errors of the mean of the data obtained are missing throughout the test
2. it is unclear if normality and equal variance tests were run to identify parametric and non-parametric data, please specify how you choose the right statistical test
3. representative pictures of the staining have not been inserted in fig 1, 2, and 3
4. Could you explain why the cell death index value of the treatment MPP+ + nicotine in the figure 3A is different from the one obtained in the figure 1B? and for the treatment MPP+ + choline between figure 2 and 3B?
5. Figure 3: please level out font size
6. Figure 4A and D : molecular weight are missing

Experimental design

Regarding the experimental design there are two main concerns for me:
• It is known that nicotine concentration in smokers’ body is below 100 µM (Matta et al., 2006; Clunes et al., 2008), so I wonder whether the concentration (2 mM) you used in your paper is appropriate.
• MLA want to be used as α7 nAChR antagonist; however at concentrations > 40 nM, the MLA purchased from Tocris (as the one used) could interacts with α4β2 and α6β2 receptors. How do you validate the concentration (20 µM) that you applied?
Methods are described with sufficient details, but I ask you to specify the mouse strain used.

To further strength Lu et al findings and correlate the in vitro part with the in vivo one, I may suggest three missing experiments:
-evaluate PARP-1 and caspase 3 expression in SH-SY5Y cells in control and after MPP, nicotine, or nicotine + MPP treatments.
-quantify cell death in the striatal tissue both in the lesioned and non-lesioned hemisphere in mice injected with nicotine or saline.
-evaluate PARP-1 and caspase 3 expression in the non-lesioned hemisphere.

Validity of the findings

It is almost impossible to evaluate data strength inasmuch means have not been inserted into the text and it is unclear if data have been correctly analyzed. Moreover, some controls are missing (PARP-1 and caspase 3 expression in the non-lesioned hemisphere).

Reviewer 2 ·

Basic reporting

Clear and concise.

Experimental design

Fine

Validity of the findings

Findings are novel and likely to be important. Links well with previous studies. Little speculation.

Additional comments

Line 301-3: I would suggest removing the following from the penultimate paragraph: “PARP-1 also plays a role in cancers, including those of the prostate and ovary, prompting the development of PARP-1 inhibitors such as olaparib as anti-cancer drugs (Deshmukh & Qiu 2015; Passeri et al. 2016).” This paragraph could be concluded by something on the importance of the sirtuin decrease arising from PARP-1 induced depletion of NAD+, with implications for mitochondrial functioning (e.g. Anderson and Maes, 2014).

It should also be noted that a7nAChR agonists are also likely to have significant impacts in PD via the regulation of e.g. the immune system and intestinal permeability (e.g. Anderson et al, 2016). This would not negate the relevance of the current results but would briefly provide the reader with a broader perspective on the role of the a7nAChR in PD.


Line 305: In the concluding paragraph, it would be better to refer to PD models and in vitro model, rather than showing “neuroprotective effects of nicotine in PD”. It is widely accepted that these commonly used PD models are poor representations of the complexity of the human condition.


Suggested references:
Anderson G, Maes M. Neurodegeneration in Parkinson's disease: interactions of oxidative stress, tryptophan catabolites and depression with mitochondria and sirtuins. Mol Neurobiol. 2014 Apr;49(2):771-83.

Anderson G, Seo M, Berk M, Carvalho AF, Maes M. Gut Permeability and Microbiota in Parkinson's Disease: Role of Depression, Tryptophan Catabolites, Oxidative and Nitrosative Stress and Melatonergic Pathways. Curr Pharm Des. 2016;22(40):6142-6151.

---

## Round 0.2 · Minor Revisions

Thank for revising your manuscript and for adding new data as requested by one of our reviewers. I will not send the paper for another round of review, but there some points that you have to clarify or correct in the revised manuscript. First, explain in the text (it can be in the legend of the figure) the main and interaction effects (in the case of use of two-way ANOVA) and also provide the F values and the degrees of freedom for them. Although you have explained that you have performed analysis of homogeneity of variance, it is not clear if the corrections made (in the case of violation of the homogeneity assumptions) was sufficient to correct the problem. Furthermore, I have not found where you have used the Dunnett's test.

In relation to figures 4, how could you perform a two-way ANOVA? The control groups have no variance and this is expected to introduce an unpredictable error in the analysis.

The authors have to clarify whether or not they have used paired t-test in the experiments with 6OHDA. Furthermore, is it appropriate to usea two-way ANOVA when the same animal is the control of the lesioned side? In this case, the measurements (dependent variables) should be treated as dependent of the subject (within subject).

---

## Round 0.3 · accepted · Accept

Thank you for correcting the manuscript. However, please check the values "F1,6= 3784, p>0.05". Something is wrong, if this unusual F value is correct, the value of P is not, etc.